# Knowledge and prevalence of common sexually transmitted infections among patients seeking care at selected health facilities in Southern Ghana

**Araba Ata Hutton-Nyameaye**[1,2]*, **Morrison Asiamah**[3], **Karikari Asafo-Adjei**[4], **Charles Kwaku Benneh**[5], **Adwoa Oforiwaa Kwakye**[6], **Kofi Boamah Mensah**[1], **Kwabena Obeng Duedu**[7,8], **Kwame Ohene Buabeng**[1,2]

1 Faculty of Pharmacy and Pharmaceutical Sciences, Department of Pharmacy Practice, Kwame Nkrumah University of Science and Technology, Kumasi, Ashanti Region, Ghana, 2 Department of Pharmacy Practice, School of Pharmacy, University of Health and Allied Sciences, Ho, Volta Region, Ghana, 3 Department of Electron Microscopy and Histopathology, Noguchi Memorial Institute for Medical Research, Accra, University of Ghana, Greater Accra Region, Ghana, 4 Microbiology Unit, Ho Teaching Hospital, Ho, Volta Region, Ghana, 5 Department of Pharmacology and Toxicology, School of Pharmacy, University of Health and Allied Sciences, Ho, Volta Region, Ghana, 6 Department of Pharmacy Practice and Clinical Pharmacy, School of Pharmacy, University of Ghana, Legon, Greater Accra Region, Ghana, 7 Department of Biomedical Sciences, School of Basic and Biomedical Sciences, University of Health and Allied Sciences, Ho, Volta Region, Ghana, 8 College of Life Sciences, Birmingham City University, City South Campus, Birmingham, United Kingdom

* aharaba@uhas.edu.gh

**Data Availability Statement:** The data has been uploaded as supporting information/supplementary file.

## Abstract

The burden of Sexually transmitted infections (STIs) remains a public health problem that should be addressed considering its effect on society and close association with HIV. This study aimed to determine the knowledge and prevalence of common STIs and associated risk factors among adult patients seeking STI care in health facilities in Ho Municipality. This was an analytical cross-sectional study involving 178 adult clients seeking treatment for suspected STIs, from November 2020 to April 2021. Data on participants' demographic characteristics, knowledge and health-seeking behaviour for STI therapy was obtained. Urine and blood samples were also taken from each participant for microbiological screening to identify the infecting pathogen and the specific STI. Multiple logistic regression and chi-square analyses were used to test the significance of associations. Of the 178 participants, 71.91% (n = 128) were women and 61.24% (n = 109) were unmarried. About 13% (n = 23) had poor knowledge of STIs. Prevalence of the STIs was 24.72% (n = 44) of which gonorrhoea was the highest 11.24% (n = 20), followed by chlamydia 10.11% (n = 18) and syphilis 7.30% (n = 13). Of all the participants, 3.37% (n = 6) had co-infections with at least 2 pathogens. Infection with all three pathogens was observed in a single participant. Participants who were married were associated with 61% reduced odds of sexually transmitted infection compared to participants who were unmarried (AOR = 0.39; CI = 0.17–0.89). Participants who smoked had 6.5 times increased odds of the infection compared to nonsmoking participants (AOR = 6.51; CI = 1.07–39.56). Although knowledge of STIs was high, it did not appear to contribute to lowering of the prevalence. This suggests there may be other factors other than

**Funding:** The authors received no specific funding for this work.

**Competing interests:** The authors have declared that no competing interests exist.

awareness or knowledge driving STIs. There is an urgent need for further studies to ascertain the drivers of STIs beyond knowledge and awareness in the public.

## Introduction

Sexually Transmitted Infections (STIs) are diseases contracted through sexual activity with a partner who has the infection [1]. Worldwide, over 1 million people are affected by STIs every day, with individuals experiencing one or more types of these infections [2]. In 2016, there was an estimated 367 million new cases of treatable STIs (chlamydia, gonorrhoea, syphilis, and trichomoniasis) globally [3]. Developing countries in regions such as South and Southeast Asia, Sub-Saharan Africa, Latin America, and the Caribbean recorded the highest proportion of STIs [4]. In underdeveloped countries, STIs are projected to be among the public health issues that impact the healthy life years of affected individuals [5]. An estimated 109.7 million individuals in Africa are currently affected by STIs, resulting in a significant 17% economic loss for the continent [6]. In Ghana, it has been approximated that 3.4% of the population is affected by STIs, and among individuals exhibiting symptoms, rates can be as elevated as 28% [7].

Sexually transmitted infections impose a significant burden of illness and death in numerous developing nations due to their impact on reproductive and infant health, as well as their contribution to the spread of Human Immunodeficiency Virus (HIV) infection. Understanding STIs and their potential repercussions is crucial for effective prevention and treatment. Individuals who lack knowledge about the symptoms may be unable to identify the urgency of seeking assistance, resulting in a failure to seek appropriate help [8]. For interventions and resources aimed at reducing the burden of STIs to be planned and implemented effectively, the prevalence and incidence have to be quantified [9].

In many parts of Africa, STIs are one of the most common reasons for seeking medical attention. Despite their broad implications, individuals' knowledge on STIs is limited [10]. Insufficient reporting regarding sexual behaviours can be caused by a lack of understanding about STIs as well as inappropriate sexual conduct [11]. In Ghana, like many other regions, the burden of STIs remains a critical concern, necessitating targeted efforts to enhance awareness, promote prevention, and improve access to quality healthcare services. This study therefore sought to identify health seeking behaviours and prevalent STIs as well as knowledge on STIs in adult patients seeking STI care in health facilities in Ho Municipality.

## Methods

### Ethical statement

Ethical approval was granted by Ghana Health Service Ethics Review Committee (GHS-ERC002/05/20), Committee for Human Research and Publication Ethics (CHRPE-KNUST) [CHRPE/AP/178/21] and Ho Teaching Hospital (HTH/RPPME/20/23) to begin data collection. Written consent was obtained from the study participants before the questionnaire was administered to them. The approved consent forms were kept securely under lock and key for review by the ethics committees. Participants were assured of privacy and confidentiality throughout the study.

### Study setting and population

Facility-based cross-sectional study was employed using a validated and well-structured, pretested, open and close ended questionnaire for data collection from patients visiting three health facilities namely Ho Teaching Hospital, Ho Municipal Hospital and Ho Polyclinic in

the Ho municipality. These three target facilities are the major health providers within the Ho Municipality.

The study population was patients visiting the Ho Teaching Hospital, Ho Municipal Hospital and Ho Polyclinic with suspected cases of STI. The average number of patients who visit these study sites were 60,000, 30,000 and 24,000 respectively annually. These health facilities offer both outpatient and inpatient department services and has functional units such as pharmacies and laboratory departments. Ho is the capital of the Volta region, which is one of the 16 regions in Ghana. It has a total area of 9,504 $km^2$ and found west to the Republic of Togo and faces the east side of Lake Volta with a population of 218,650 [12].

Patients between the ages of 18 years to 65 years who visited the three hospitals with STIs and willing to participate in the study were included. The participants' symptoms included lower abdominal pain (females only), pain on urination, genital sores, and/or discharges from the penis or vagina. The exclusion criteria were persons who are unable to carry out self-care or receive medication by themselves.

The determination of the sample size was calculated using the Cochran formula, with prevalence assumed to be 7.7% based on a study conducted by Semwogerere et al., 2021 [13] with a confidence interval of 95%, a precision of 5% and non-response rate of 20%. The minimum number of 131 was required for the study, but the final sample size was 178.

## Data collection process

Questionnaires were administered to patients with suspected cases of STI (study participants) seeking care and treatment at the study sites willing to participate in the study. Physicians at the various health facilities introduced the study to patients who had been diagnosed syndromically with STI after consultation. Syndromic care of STIs involves making a preliminary diagnosis by observing a specific collection of signs and symptoms in the patient [14]. The common clinical syndromes of STIs include urethral discharge in males, vaginal discharge, ano-genital ulcer disease, lower abdominal pain, skin rash–genital or generalized, scrotal swellings and genital warts [15,16]. The patients were briefed about the purpose of the study, the methods and ethical issues and those who consented to the study were included. Simple random sampling method was used to recruit the patients. The sampling method was employed to ensure every patient had an equal opportunity to take part in the study, minimize potential biases, and achieve a thorough representation of the study population. Participants were made to pick from an envelope containing folded shuffled papers with 'Yes' or 'No' written on them. Where a participant picked 'Yes' the participant was included and where the participant picked 'No' the participant was excluded.

A questionnaire was designed to collect quantitative data on participants' knowledge on STI, patient's history of STI, patient factors (smoking and alcohol intake status), sexual and health seeking behaviour. Some terms such as, "sexual partners" meant number of sexual partners the participant currently had, "past relationships" signified the number of relationships in the past year and all participants were questioned on the use of condoms and number of sexual intercourses/episodes per week. Furthermore, participants current use of alcohol was assessed while participants with/without knowledge on STI was asked if they had "heard of STI". Also, orthodox treatment generally applied to the proven and widely acknowledged medical interventions that are practiced within the healthcare system. The questionnaire was developed from existing literature after careful review of literature.

## Quality control

To assure the accuracy of the data, the validity and dependability of the study instrument were validated beforehand. A pilot test was also done with ten (10) patients in order to find and

remove all errors as well as discrepancies within the questionnaire and authenticate it. Three (3) experienced research assistants in STIs were trained to aid with the gathering of data from the participants. Before the participants began answering, they were provided with a thorough explanation of the questions.

## Laboratory analysis

Microbiological screening was done by collecting blood and urine samples from each participant to objectively assess the prevalence of the common STIs in the study area. Syphilis test kits (Wondfo One Step Syphilis- Sensitivity:100% and Specificity:98.8%)) and Cepheid Xpert CT/NG (Female urine- CT/NG Sensitivity:98.1% / 94.4%, Specificity:99.8% />99.9%, PPV [99.8%] / NPV [100%]; Male urine- CT/NG Sensitivity:98.5%/ 98.3%, Specificity:99.8% / 99.9%, PPV [99.8%]/ NPV [100%]) assay on the GeneXpert systems were used for detecting the presence of syphilis, chlamydia, and gonorrhoea respectively. As syphilis testing involved only treponemal antibodies, it is unable to differentiate between active verses past infection. All samples were collected and analysed according to the manufacturers' protocols.

## Statistical analysis

Descriptive statistics were used to summarize the socio-demographic characteristics of the study participants. Categorical predictor variables were summarized using frequencies and percentages, while continuous predictor variables were done using means and medians. The proportion diagnosed of STI was estimated as the number of participants who tested positive to any of the STI tests divided by the number of participants who were tested.

Also, the level of knowledge was assessed as a 24-item composite variable. Each item of the variable was assigned a score of "0" if the response to the item was wrong and a score of "1" if a response to an item was correct. The maximum score a participant could obtain was 24 points and the minimum score a participant could obtain was zero. Participants who scored a total of 0 to 8 points were considered as poor knowledge, 9 to 16 points as satisfactory knowledge and 17 to 24 points as good knowledge.

The associations between demographic characteristics, knowledge and risk of contracting an STI as well as behavioural factors were determined using inferential statistics. Factors that influenced the risk of STI were initially assessed by conducting a bivariate analysis using Pearson's chi square test, then independent variable significant under this test were included in the multiple logistic regression to estimate the adjusted strength of association between the risk of STI and the predictors. STATA 15 was used for the analysis. The strength of association was reported as odd ratios. Associations with p-values of less than 0.05 were considered statistically significant.

## Results

### Demographic characteristics of the participants

The number of study participants was one hundred and seventy-eight (178). The mean age of the study participants was 28yrs ± 0.64 with the majority being between 24–40 years (n = 94, 52.81%). There were more females (n = 128, 71.91%) than males. About half of the participants reported to the Ho Municipal Hospital (n = 90, 50.56%). Almost all participants were handled at the outpatient department (OPD) (94.38%) with the exception of a few who were handled at the Obstetrics and Gynecology Unit. The detailed demographic characteristics of the study population is presented in Table 1 below.

**Table 1. Demographic characteristics of participants.**

| Variable | Category | Frequency | Percentages |
|---|---|---|---|
| Age (years) | Less than 24 | 66 | 37.08 |
| | 24 to 40 | 94 | 52.81 |
| | More than 40 | 18 | 10.11 |
| Name of hospital | Ho Teaching Hospital | 53 | 29.78 |
| | Ho Polyclinic | 35 | 19.66 |
| | Ho Municipal Hospital | 90 | 50.56 |
| Occupation | Unemployed | 38 | 21.59 |
| | Employed | 138 | 78.41 |
| Unit | OPD | 168 | 94.38 |
| | Obstetrics and Gynae | 10 | 5.61 |
| Gender | Male | 50 | 28.09 |
| | Female | 128 | 71.91 |
| Education | No Education | 4 | 2.25 |
| | Primary Education | 36 | 20.22 |
| | Secondary Education | 73 | 41.01 |
| | Tertiary Education | 65 | 36.52 |
| Marital Status | Unmarried | 109 | 62.64 |
| | Married | 65 | 37.36 |
| Religion | Christian | 175 | 98.31 |
| | Muslim | 3 | 1.69 |
| Tested for STI | Negative | 134 | 75.28 |
| | Positive | 44 | 24.72 |

## Symptoms exhibited by participants

The most common symptoms reported by men and women were discharge from penis (urethral discharge) [n = 38,76%] and vaginal discharge (n = 109,85.2%) respectively (Table 2).

## Prevalence and knowledge of STIs among study participants

The overall prevalence of STIs was 24.72% (Table 1). Of these 8.43% (n = 15), 7.30% (n = 13) and 5.62% (n = 10) were gonorrhea only, chlamydial infections only, and syphilis only respectively (Table 3). Low levels of knowledge did not translate to a higher number of infections (Table 3). Among the participants, 3.37% (n = 6) had co-infections with two or three pathogens. Co-infections involving gonorrhoea and chlamydia were 1.69% (n = 3). Other dual co-

**Table 2. Symptoms reported by participants.**

| Symptoms | Category | Gender | | Frequency (Percentages) |
|---|---|---|---|---|
| | | Male | Female | |
| Discharge from Genitalia | Present | 38 (76.0%) | 109 (85.2%) | 147 (82.6%) |
| | Absent | 12 (24.0%) | 19 (14.8%) | 31 (17.42%) |
| Pain on Urination | Present | 27 (54.0%) | 29 (22.7%) | 56 (31.5%) |
| | Absent | 23 (46.0%) | 99 (77.3%) | 122 (68.5%) |
| Lower abdominal pain | Present | 2 (4.0%) | 43 (33.6%) | 45 (25.3%) |
| | Absent | 48 (96.0%) | 85 (66.4%) | 133 (74.7%) |
| Genital Sores | Present | 4 (8.0%) | 11 (8.6%) | 15 (8.4%) |
| | Absent | 46 (92.0%) | 117 (91.4%) | 163 (91.6%) |

**Table 3. Prevalence of STIs according to level of knowledge.**

| Infections | Level of Knowledge | | | | Total (%) |
|---|---|---|---|---|---|
| | Poor | Satisfactory | Good | Missing | |
| Chlamydia only | 2 (4.55) | 4 (9.09) | 7 (15.91) | 0 (0.00) | 13 (29.55) |
| Gonorrhoea only | 3 (6.82) | 9 (20.45) | 3 (6.82) | 0 (0.00) | 15 (34.09) |
| Syphilis only | 2 (4.55) | 4 (9.09) | 4 (9.09) | 0 (0.00) | 10 (22.73) |
| Co-infections | 1 (2.27) | 3 (6.82) | 1 (2.27) | 1 (2.27) | 6 (13.63) |
| **Total** | **8 (18.18)** | **20 (45.45)** | **15 (34.09)** | **1 (2.27)** | **44 (100)** |

infections as well as co-infections with all three pathogens were recorded in one participant each.

## Association between STI status, demographic and behavourial characteristics

Under the analysis, gender and marital status of the participants were statistically associated with the risk of sexually transmitted infection among the respondents. Additionally, females were 60% less likely to be at risk of STI infection compared to males (COR = 0.40; CI = 0.19–0.81). Married participants were 58% less likely to be at risk of STI compared to unmarried participants (COR = 0.42; CI = 0.19–0.93) (Table 4).

Smokers were 8.5 times at risk of STI infection compared to non smokers (COR = 8.46; CI = 0.58–45.32) and having two or more past relationships increased risk of STI by two folds compared to having less than two past relationships (COR = 2.32; CI = 1.11–4.84). Other demographic and behavioral characteristics such as age, study site, occupation, educational level, information on STI, alcohol consumption, number of sexual partners, condom use, frequency of sexual intercourse and preferred treatment facility were not statistically associated with risk of sexually transmitted infection (Table 4).

From the multivariable analysis using the multiple logistic regression, the factors that were found to influence the risk of sexually transmitted infection were the marital and smoking statuses of the participants. It was observed that being married was associated with 61% reduced odds of sexually transmitted infection compared to being unmarried (AOR = 0.39; Cl 0.17–0.89). Participants who smoked were 6.5 times more likely to contract an STI (AOR = 6.51; Cl 1.07–39.56) (Table 5).

## Health-seeking behaviour of participants

Self-medication was reported by 44.07% (n = 78) of the participants. Regarding where participants would normally prefer to seek care, if need be, 87.08% (n = 155) preferred to seek care in hospitals. Regarding the choice of treatment type, 79.21% (n = 141) preferred to orthodox treatment as opposed to herbal treatment. About 19.32% (n = 34), 24.43% (n = 43), 30.68% (n = 54) and 25.57% (n = 45) of participants reported having waited for less than 1 week, 1 to 2 weeks, 2 weeks to 1 month and more than a month respectively prior to seeking care.

## Discussion

Discharges from the penis or vagina were the most common clinical symptoms reported by the study participants and this is consistent with studies done in Ghana [17,18] and South Africa [19]. The study revealed a concerning overall prevalence of STIs at 24.72%, highlighting a significant public health issue. Self-reported prevalence of symptomatic STI among female sex workers in Nigeria in a similar study by Sekoni et al [20] was 36.5%. In addition, similar

**Table 4. Strength of association between STI status, demographic and behavourial characteristics.**

| Variable | Category | STI | | COR | (95%CI) |
|---|---|---|---|---|---|
| | | **Negative** | **Positive** | | |
| Age | Less than 24 | 45 (68.2%) | 21 (31.8) | Ref | |
| | 24 to 40 | 74 (78.7) | 20 (21.3) | 0.58 | 0.28–1.18 |
| | More than 40 | 15 (83.3) | 3 (16.7) | 0.43 | 0.11–1.64 |
| Study Site | Ho Teaching Hospital | 38 (71.7%) | 15 (28.3%) | Ref | |
| | Ho Polyclinic | 29 (82.9%) | 6 (17.1%) | 0.52 | 0.18–1.52 |
| | Ho Municipal Hospital | 67 (74.4%) | 23 (25.6%) | 0.87 | 0.41–1.86 |
| Occupation | Unemployed | 26 (68.4%) | 12 (31.6%) | Ref | |
| | Employed | 106 (76.8%) | 32 (23.2%) | 0.65 | 0.30–1.44 |
| Gender | Male | 31 (62.0%) | 19 (38.0%) | Ref | |
| | Female | 103 (80.5%) | 25 (19.5%) | **0.40** | **0.19–0.81*** |
| Education | No formal/ Primary educ. | 31(77.5%) | 9 (22.5%) | Ref | |
| | Secondary Education | 53 (72.6%) | 20 (27.4%) | 1.30 | 0.52–3.21 |
| | Tertiary Education | 50 (76.9%)) | 15 (23.1%) | 1.03 | 0.40–2.65 |
| Marital Status | Unmarried | 79 (69.9%) | 34 (30.1%) | Ref | |
| | Married | 55 (84.6%) | 10 (15.4%) | **0.42** | **0.19–0.93*** |
| Heard of STI | No | 12 (63.2%) | 7 (36.8%) | Ref | |
| | Yes | 122 (76.7%) | 37 (23.3%) | 0.52 | 0.19–1.42 |
| Level of Knowledge | Poor | 15 (65.2%) | 8 (34.8%) | Ref | |
| | Satisfactory | 77 (79.4%) | 20 (20.6%) | 0.49 | 0.18–1.31 |
| | Good | 42 (73.7%) | 15 (26.3%) | 0.67 | 0.24–1.90 |
| Drink Alcohol | No | 97 (79.5%) | 25 (20.5%) | Ref | |
| | Yes | 37 (66.1%) | 19 (33.9%) | 1.99 | 0.99–4.04 |
| Smoking Status | Non-Smoker | 132 (77.2%) | 39 (22.8%) | Ref | |
| | Smoker | 2 (28.6%) | 5 (71.4%) | **8.46** | **1.58–45.32*** |
| Sexual Partners | No sexual partner | 16 (69.6%) | 7 (30.4%) | Ref | |
| | One | 102 (77.3%) | 30 (22.7%) | 0.67 | 0.25–1.79 |
| | Two or more | 13 (65.0%) | 7 (35.0%) | 1.23 | 0.34–4.42 |
| Condom use | No | 81 (76.4%) | 25 (23.6%) | Ref | |
| | Yes | 53 (73.6%) | 19 (26.4%) | 1.16 | 0.58–2.32 |
| Preferred treatment facility | Comm. pharmacy | 19 (82.6%) | 4 (17.4%) | Ref | |
| | Hospital | 115 (74.2%) | 40 (25.8%) | 1.65 | 0.53–5.15 |
| Sexual intercourse per week | Less than 3 | 66 (74.2%) | 23 (25.8%) | Ref | |
| | 3–4 | 53 (80.3%) | 13 (19.7%) | 0.70 | 0.33–1.52 |
| | Greater than 4 | 6 (66.7%) | 3 (33.3%) | 1.43 | 0.33–6.21 |
| Past Relationships | one or less | 103 (79.2%) | 27 (20.8%) | Ref | |
| | 2 or more | 28 (62.2%) | 17 (37.8%) | **2.32** | **1.11–4.84*** |

studies in Ethiopia (Gondar) and Gambia by Geremew et al [21] and Butcher et al [22] reported prevalence of STI to be 74.1% and 9.8% respectively.

Further analysis of the cohort used in this study, indicated that gonorrhea, chlamydial, and syphilis infections account for 11.24%, 10.11% and 7.30% of the cases, respectively. Banong-le et al. [23] found the prevalence of syphilis infection among symptomatic patients in a study conducted in Ghana to be 3.2%. This is lower than this study findings (7.30%) [23]. Geremew et al. [21] also reported very high prevalence of syphilis (30%) and *N. gonorrhoeae* (*20.8%)* in a study conducted among symptomatic patients attending Gondar town Hospitals and Health

**Table 5. Strength of association between Risk of STI infection, gender of participant, smoking status and history of relationships.**

| Variable | Category | COR | CI | AOR | CI |
|---|---|---|---|---|---|
| Gender | Male | Ref | | Ref | |
| | Female | 0.40 | 0.19–0.81 | 0.54 | 0.24–1.22 |
| Marital Status | Unmarried | Ref | | Ref | |
| | Married | 0.42 | 0.19–0.93 | **0.39** | **0.17–0.89*** |
| Smoking Status | Non-Smoker | Ref | | Ref | |
| | Smoker | 8.46 | 1.58–45.32 | **6.51** | **1.07–39.56*** |
| Past Relationships | one or less | Ref | | Ref | |
| | 2 or more | 2.32 | 1.11–4.84 | 6.51 | 0.76–3.78 |

Centers [21]. The prevalence of chlamydia among patients showing symptoms in a study carried out by Nyarko et al.[17] in Western Ghana was 20.4% and this is higher than chlamydia prevalence in this current study. These findings may due to difference in study population, study facilities and study location.

The study brings attention to the complexity of STI cases, with 3.37% of participants experiencing co-infections involving two or three pathogens. Co-infections of gonorrhea and chlamydia were observed in 1.69% of cases, while other dual co-infections and co-infections with all three pathogens were less common but still present. These findings emphasize the importance of comprehensive STI screening and management strategies to address the multifaceted nature of STI transmission. Generally, while laboratory diagnosis of STIs is more reliable, it is also time-consuming, expensive, and involves advanced technology and resources. This makes it challenging to utilise regularly in countries with limited resources. The majority of countries face a significant burden of STIs; nonetheless, they do not have the requisite technical expertise, specialised doctors, and laboratory infrastructure to effectively diagnose these STIs [24]. Considering the increased risk of antibiotic resistance and complications with untreated STI, clinicians should increase diagnostic testing of participants who visit their health facilities with STI syndromes. This would help identify the specific type of STI before treatment is started.

Interestingly, the results of this study suggests that low levels of knowledge about STIs did not necessarily correlate with a higher incidence of infections, as indicated in Table 3. Individuals encounter numerous obstacles in their efforts to prevent STIs, including discrimination, inadequate healthcare systems, insufficient money, and lack of legislative backing. The presence of social stigma related to STIs, especially within certain populations such as men who have sex with men (MSM), has a substantial influence on people's inclination to seek medical assistance or reveal their sexual behaviour while seeking healthcare [25,26]. The findings in this study also underscores the need for targeted educational campaigns and interventions to bridge the gap between awareness and behaviours.

Married individuals were found to be 61% less likely to face STI risk in contrast to unmarried participants and this is in concordance with a study conducted by Taylor et al. [27] which signified that the occurrence of multiple partners and long-lasting relationships within a year was minimal among married individuals, moderate among cohabiting couples, and highest among individuals who were previously married or had never been married [27].

Additionally, smokers were identified as 6.5 times more susceptible to STI infection compared to nonsmokers. Even though there was a total of only seven (7) smokers in the whole cohort, the 95% confidence interval for the odds of STI for smokers versus non-smokers was very wide in multivariable analysis (1.07–39.56). This emphasized the association between smoking and its heightened susceptibility to STIs, highlighting the importance of targeted

interventions for this population. Studies done by Berg et al.[28] and Bajaj et al.[29] in America and Canada respectively established that smoking is associated with a person's tendency for dangerous sexual behaviour and raised the probability of developing an STI.

The findings on marital status and smoking status highlight the nuanced interplay of demographic and behavioral factors in shaping STI vulnerability, emphasizing the need for targeted interventions and public health initiatives to address specific risk factors identified in this study.

Considering the health-seeking behaviour of participants towards STI management in this study, this was very alarming because patients delayed in seeking healthcare for their genitourinary symptoms. It is recommended that patients with genitourinary symptoms must seek care immediately to prevent complications such as infertility. The possible reason for the delay maybe due to the cultural sensitiveness of discussing genitourinary issues in the Ghanaian setting [30]. Secondly, from the health belief model, the delay in seeking care questioned the confidence the patients had in the overall health system. Additionally, patients may have sought care outside the health system and also, the high cost of treatment. Delayed treatment is a bane to the effective management of STI as it may lead to Pelvic Inflammatory Diseases (PID), etc [31].

Some of the patients claimed that they had taken antimicrobials, antihistamines and herbal concoctions prior to seeking healthcare at the hospital. Self-medication poses a lot of challenges to the management of medical conditions even though STI patients are more likely to self-medicate on antibiotics [32]. This leads to an increase in antimicrobial resistance and its consequences.

The hospital is thus the recommended point of care as reported in this study. For community pharmacies and herbal centers, they may not be trained to adequately diagnose the aetiology of these symptoms. These syndromes of STIs could lead to a reduced quality of life of the individual [33] with inappropriate drug therapy. Patient's preference is an integral factor that has been established to improving compliance to treatment [34] and this study finding showed that patients preferred both orthodox and herbal treatment although there is paucity of evidence on the efficacy and safety of herbal medicines on the management of STIs. This highlights the need for research on the safety and efficacy of herbal medicines on the Ghanaian market so that policies can focus on incorporating safe and efficacious herbal medicines into the national treatment guidelines.

One of the strengths of this study included the use of very sensitive and specific approved kits to detect the presence of multiple infectious organisms which overall improves the validity of the study findings.

The limitations of this study were that the study focused on only three common bacterial STIs even though all patients who presented with suspected cases of all types of STIs were included. Thus, there is an increased chance that the burden of STI would be higher than what we estimated considering other STIs such as chancroid, HIV/AIDs, herpes, etc. This study employed a cross-sectional design hence the conclusions are limited to associations not causality. This study is also subject to recall bias as participants were made to remember past events and hence there is a potential for inaccurate responses due to recall bias. To reduce the impact of recall bias, we reduced the number of items on the questionnaire that required recall. Additionally, the knowledge assessment of STI was not validated and cannot be generalized. It is difficult to make associations between STI knowledge and STI prevalence among those already symptomatic. Even those with low knowledge were at the health facilities because they were seeking care. There could still be an association between STI knowledge and STI prevalence among people with asymptomatic infections.

## Conclusion

Almost thirty percent of the study participants had STI. The most vulnerable age group was 24–40 years. Gonorrhoea, chlamydia and syphilis and associated co-infections were prevalent in the participants seeking medical attention with suspected cases of STIs. Also, low STI knowledge was not associated with having an STI in this study. Demographic and behavioural characteristics such as marital status and smoking were found to be associated with the risk of contracting an STI. Continuous awareness and sensitization about STIs are essential to reducing the risk and spread of the problem. The prevalence of STI among patients with the suspected illness calls for more interventions to improve targeted treatment for better outcomes and minimize risks for antimicrobial resistance. The Government policy on STI education should therefore be reevaluated and more tangible and long-lasting strategies devised to strengthen knowledge and awareness about the mentioned STIs in this study and others reported in the Ghanaian health system to comprehensively combat these infections including HIV/AIDs.

## Supporting information

**S1 Questionnaire. QUESTIONNAIRE.**
(DOCX)

**S1 Data.**
(DTA)

## Acknowledgments

We thank the management, staff and study participants of Ho Teaching Hospital, Ho Polyclinic and Ho Municipal Hospital for their support during the study.

## Author Contributions

**Conceptualization:** Araba Ata Hutton-Nyameaye, Morrison Asiamah, Charles Kwaku Benneh, Kofi Boamah Mensah, Kwabena Obeng Duedu, Kwame Ohene Buabeng.

**Data curation:** Araba Ata Hutton-Nyameaye, Karikari Asafo-Adjei.

**Formal analysis:** Araba Ata Hutton-Nyameaye, Morrison Asiamah, Kwabena Obeng Duedu.

**Funding acquisition:** Araba Ata Hutton-Nyameaye.

**Investigation:** Araba Ata Hutton-Nyameaye, Karikari Asafo-Adjei.

**Methodology:** Araba Ata Hutton-Nyameaye, Morrison Asiamah, Karikari Asafo-Adjei, Adwoa Oforiwaa Kwakye, Kwame Ohene Buabeng.

**Project administration:** Araba Ata Hutton-Nyameaye.

**Resources:** Araba Ata Hutton-Nyameaye.

**Software:** Morrison Asiamah.

**Supervision:** Kwabena Obeng Duedu, Kwame Ohene Buabeng.

**Validation:** Araba Ata Hutton-Nyameaye, Morrison Asiamah, Charles Kwaku Benneh.

**Visualization:** Araba Ata Hutton-Nyameaye, Morrison Asiamah, Charles Kwaku Benneh, Adwoa Oforiwaa Kwakye, Kwabena Obeng Duedu.

**Writing – original draft:** Araba Ata Hutton-Nyameaye, Morrison Asiamah, Karikari Asafo-Adjei, Charles Kwaku Benneh, Adwoa Oforiwaa Kwakye, Kofi Boamah Mensah.

**Writing – review & editing:** Araba Ata Hutton-Nyameaye, Morrison Asiamah, Karikari Asafo-Adjei, Charles Kwaku Benneh, Adwoa Oforiwaa Kwakye, Kofi Boamah Mensah, Kwabena Obeng Duedu, Kwame Ohene Buabeng.

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
