## [Decision Letter · Decision Letter 0]

25 Sep 2023

PGPH-D-23-01356

Knowledge and prevalence of common sexually transmitted infections among patients seeking care at selected health facilities in Southern Ghana

Dear Dr. Hutton-Nyameaye,

Thank you for submitting your manuscript to PLOS Global Public Health. After careful consideration, we feel that it has merit but does not fully meet PLOS Global Public Health’s publication criteria as it currently stands. Therefore, we invite you to submit a revised version of the manuscript that addresses the points raised during the review process.

Please see the comments from three reviewers below. Reviewer 2 in particular has expressed significant concerns about several aspects of the manuscript and reporting, including interpretation of previous work. We now invite you to address these concerns.

We look forward to receiving your revised manuscript.

Kind regards,

Hanna Landenmark

Staff Editor

Journal Requirements:

1. Please provide separate figure files in .tif or .eps format only and remove any figures embedded in your manuscript file. Please also ensure all files are under our size limit of 10MB.

2. We do not publish any copyright or trademark symbols that usually accompany proprietary names, eg  ©, ®, ™  (e.g. next to drug or reagent names). Please remove all instances of trademark/copyright symbols throughout the text, including ® on page 9.

3. In the online submission form, you indicated that "The data is available upon reasonable request". All PLOS journals now require all data underlying the findings described in their manuscript to be freely available to other researchers, either 1. In a public repository, 2. Within the manuscript itself, or 3. Uploaded as supplementary information.

Additional Editor Comments (if provided):

Reviewers' comments:

Reviewer's Responses to Questions

**Comments to the Author**

1. Does this manuscript meet PLOS Global Public Health’s publication criteria? Is the manuscript technically sound, and do the data support the conclusions? The manuscript must describe methodologically and ethically rigorous research with conclusions that are appropriately drawn based on the data presented.

Reviewer #1: Yes

Reviewer #2: No

Reviewer #3: Yes

2. Has the statistical analysis been performed appropriately and rigorously?

Reviewer #1: No

Reviewer #2: Yes

Reviewer #3: Yes

3. Have the authors made all data underlying the findings in their manuscript fully available (please refer to the Data Availability Statement at the start of the manuscript PDF file)?

Reviewer #1: Yes

Reviewer #2: No

Reviewer #3: Yes

4. Is the manuscript presented in an intelligible fashion and written in standard English?

Reviewer #1: Yes

Reviewer #2: Yes

Reviewer #3: Yes

5. Review Comments to the Author

Reviewer #1: Thank you for the opportunity to review the manuscript on "Knowledge and prevalence of common sexually transmitted infections among patients seeking care at selected health facilities in Southern Ghana"

This is a valuable contribution to public health and to the body of knowledge on STIs control and management.

I am satisfied with other sections of the manuscript but there is a need for the authors to revisit data management, data analysis, interpretations and discussion of the results to improve the manuscript.

Data management - data needs to be prepare for reanalysis

-Some variables have many subcategories and it is not necessary.

Age categories: has five categories Less than 21, 21 - 30, 31-40; 41 - 50

Greater than 50. I advise the authors to collapse to at least 3 categories for the association to make sense. EG. 18-24, 25-40, 41 and above.

Level of Knowledge =Poor, Satisfactory, Good, Excellent. Authors can group excellent and good to remain with three categories

No sexual partners has FIVE categories. I suggest 3 categories no partner, one , two or more

Past relationships has SIX categories. I suggest 3 no relationship, one, two or more

Regrouping categories will improve the results when computing the association and strength of association, especially because some values for subcategories = 1, which dilutes the analysis.

Occupation is also a long list that need regrouping

Results

When presenting association and strength of association, the authors do not give clear interpretation of the results by subgroup? EG: LINE 171-184 please clarify which SUBCATEGORY is associated with to risk

Youth, average age or older age? male or female? smoker on nonsmoker, few or more sexual partners? few or more past relationships, low or high level of knowledge etc. This results will be improved once the subcategories are collapsed as suggested in the data management.

Discussion

For all the statement made in the discussion, the authors cite only one study. This show lack of synthesis and balanced comparison with other studies. I encourage the authors to include more studies because STIs is a global topic and there are a lot of studies even in the region of Africa to compare with.

Abstract

After collapse subcategories and running the statistics, the authors have to realign the assignment.

Reviewer #2: Overall

This manuscript presents data from a cross-sectional study among 178 individuals seeking STI care in Ho Municipality in Ghana. Prevalence of chlamydia, gonorrhoea, and syphilis, in this population was 35.3%, 39.2%, and 25.5%, respectively. This is likely to be expected given they are reported to be symptomatic. Multivariate logistic regression is used to assess factors associated with STI diagnosis.

The introduction is very broad, and doesn’t provide enough background information to justify the study. If data on current prevalence of STIs in Ghana is available, please provide it. If no such data is available, state this. Furthermore, a large sub-section of the study is based around a single study by Seidu et al., which demonstrated self-reporting of STIs in the previous 12 months from demographic health and surveillance data. The results of the study appear to be misrepresented, and so this section needs to be re-formulated.

The methods are relatively clear, and I have just suggested a few minor changes for clarity. However, a significant issue is the lack of symptoms data as it makes it more difficult to determine who the sample is and who they are meant to represent.

Some aspects of the results section are unclear, particularly surrounding presentation of data.

The discussion needs a significant re-writing. There are several errors related to presenting this study’s own results. Furthermore, some of the comparisons with other studies and data are inappropriate due to differences in study design and populations. There needs to a unifying focus or thread through the discussion – at present, it is quite long but very disjointed, jumping from one topic to the next.

Overall, there are several major issues that need to be addressed before being suitable for publication.

Abstract

Page 2, line 25: “STIs”

1. Please provide full term before first use of abbreviations.

Page 2, lines 26-28: “This study aimed to determine the knowledge and prevalence of common STIs and associated risk factors among adult patients seeking care in health facilities in Ho Municipality.”

2. To ensure the study aim is as accurate as possible, I suggest making it explicit that the sample is adult patients seeking “STI” care.

3. Please provide numbers alongside the percentages for demographic factors and STI prevalence data.

Introduction

Page 3, lines 50-51: “Sexually Transmitted infections (STIs) are infections acquired through sexual intercourse with an infected partner (1).”

4. Please consider re-phrasing or removing the term “infected partner”, as the term “infected” is potentially stigmatising.

Page 3, lines 51-52: “Globally, STIs affect more than 1 million people worldwide, with one or more infection types occurring per day.”

5. I think this sentence is trying to convey that recent data suggests that there are around 1 million new infections of chlamydia, gonorrhoea, trichomoniasis or syphilis per day. This sentence needs to be re-phrased to make this more clear, as currently the meaning is quite confusing. Additionally, specifying the STIs that are being referred to is important. A reference is also required (current reference number 5 would be sufficient). Given that mention of 367 million new cases per year is subsequently discussed in this paragraph, consider removing this sentence altogether.

Page 3, lines 52-53: “Almost 30 different STIs have been established; some of which are easily managed while others are not treatable (2).

6. The reference for this statement is a 2013 therapeutics textbook. I suggest providing a more recent and relevant reference. Consider removing the specific number of STIs that have been “established” as this is likely to change and is open to debate.

Page 3, lined 58-60: “For interventions and resources aimed at reducing the burden of STIs to be planned and implemented effectively, the prevalence and incidence have to be quantified (6).”

7. The introduction needs to address why this specific study is required. For example, what data is already available on STIs in Ghana? What could then be done with the data? At present, the introduction is very broad.

Page 3, lines 61-62: “In many parts of Africa, STIs are one of the most common reasons for seeking medical attention.”

8. Please provide a reference to support this statement

Page 3, lines 65-67: “The prevalence of STI self-reporting was 6% in a study conducted by Seidu et al.(9) in Ghana (9). This means that a small number of patients self-report as compared to the number of patients who are clinically diagnosed with STIs.”

9. The comparison drawn is not reflective of the findings of the study by Seidu et al, which looks at the prevalence of STI self-reporting in Ghana demographic and health surveillance data. Men were asked during the survey “if they had contracted a disease through sexual intercourse in the previous 12 months”, to which they answered yes or no. No mention appears to be made in the study, comparing their findings of self-reporting with “patients who are clinically diagnosed with STIs”. Either provide an additional reference to support this comparison or change above. Additionally, the reference number “9” is in two places in the above sentence.

Page 3, lines 68-71: “Therefore, self-reporting must be encouraged to facilitate the prompt of these infections. Effective public health interventions such as self-reporting are however hinged on understanding the health seeking triggers in persons who may have STIs.”

10. Although seeking healthcare when symptomatic should be encouraged, the use of “self-reporting” here is confusing. It appears to be building on the above reference to Seidu et al. But in that study “self-reporting” did not refer to healthcare seeking – it solely referred to whether a participant reported having an STI in the previous 12 months” during a DHS survey. I think this section needs to be re-written with a better representation of what the data in Seidu et al. is presenting. Additionally, I think a word is missing after “prompt”.

Page 3, lines 73-75: “This study therefore sought to identify health seeking behaviours and prevalent STIs as well as knowledge on STIs in persons who were reporting in various health facilities in southern Ghana.”

11. As with comment above, please ensure the study aim is as accurate as possible, by making it explicit that the sample is adult patients seeking STI care.

Methods

Page 6, lines 101-103: “The deter

---

## [Decision Letter · Decision Letter 1]

19 Feb 2024

PGPH-D-23-01356R1

Knowledge and prevalence of common sexually transmitted infections among patients seeking care at selected health facilities in Southern Ghana

Dear Dr. Hutton-Nyameaye,

Thank you for submitting your manuscript to PLOS Global Public Health. After careful consideration, we feel that it has merit but does not fully meet PLOS Global Public Health’s publication criteria as it currently stands. Therefore, we invite you to submit a revised version of the manuscript that addresses the points raised during the review process.

Please pay particular attention to the comments from Reviewer 4, who requests revision of the results section for clarity regarding the total individuals included in the study, as well as clarification regarding methodological reporting and whether STI diagnosis followed routine standard of care procedures.

We look forward to receiving your revised manuscript.

Kind regards,

Jennifer Tucker, PhD

Associate Editor

Journal Requirements:

Additional Editor Comments (if provided):

Reviewers' comments:

Reviewer's Responses to Questions

**Comments to the Author**

1. If the authors have adequately addressed your comments raised in a previous round of review and you feel that this manuscript is now acceptable for publication, you may indicate that here to bypass the “Comments to the Author” section, enter your conflict of interest statement in the “Confidential to Editor” section, and submit your "Accept" recommendation.

Reviewer #4: (No Response)

Reviewer #5: All comments have been addressed

2. Does this manuscript meet PLOS Global Public Health’s publication criteria? Is the manuscript technically sound, and do the data support the conclusions? The manuscript must describe methodologically and ethically rigorous research with conclusions that are appropriately drawn based on the data presented.

Reviewer #4: Partly

Reviewer #5: Yes

3. Has the statistical analysis been performed appropriately and rigorously?

Reviewer #4: Yes

Reviewer #5: Yes

4. Have the authors made all data underlying the findings in their manuscript fully available (please refer to the Data Availability Statement at the start of the manuscript PDF file)?

Reviewer #4: Yes

Reviewer #5: Yes

5. Is the manuscript presented in an intelligible fashion and written in standard English?

Reviewer #4: Yes

Reviewer #5: Yes

6. Review Comments to the Author

Reviewer #4: This is a cross-sectional study conducted in three outpatient clinics in Ghana enrolling adults with STI symptoms to evaluate STI knowledge and the prevalence of gonorrhea, chlamydia, and syphilis. The authors found a high prevalence of curable STIs and several demographic and behavioral factors associated with having an STI.

While the previous reviewers’ comments have been addressed, I have still identified a number of items that I believe need to be rectified prior to publication, in particular a discrepancy in the total number of individuals with an STI accounting for co-infections vs the total numbers of individual STIs added together (see comment in results section with asterisks below).

Introduction

The sentence in lines 53-54 needs clarification to states that 1 million people worldwide acquire a new infection every day, not that 1 million people worldwide are affected by STIs.

In line 71, I presume that it is ‘symptoms of STIs’ that are one fo the most common reasons for seeking medical attention. If so, please clarify here.

Does the statement ‘knowledge on STIs is limited’ in line 72 refer to individuals’ knowledge of STIs and their symptoms, or knowledge about the epidemiology of STIs in Ghana? Or both? It would be helpful to clarify here, and if it is the latter (epidemiology), to expand on why. E.g. inadequate epidemiologic surveillance, lack of access to etiologic testing, etc.

Methods

Line 112 “physician at the various health facilities introduced the study to patients. Who had been diagnosed with STI after consultation” – can you please clarify how STIs are diagnosed normally at the clinical sites. Presumably this is syndromic management but it would be helpful to clarify for readers who may not be familiar with the local standard of care.

Line 121 – was any of the data collected using the questionnaire qualitative? I believe this should be "quantitative"

Laboratory analysis: the statement in lines 148-150 regarding syphilis testing only involving treponemal antibodies would be more logical to include in the laboratory analysis, rather than the statistical analysis section; recommend moving this up.

It would be very helpful to include the list of questions used to assess STI knowledge as part of this manuscript, potentially as a supplementary file, as this would be informative to readers in interpreting the results of the knowledge assesmsnet. Were these questions part of a validated STI knowledge tool used previously in other settings? What do the questions ask about - STI symptoms? transmission? sequelae? How were the score categories of 0-6, 7-12, and 13-24 points to represent low, satisfactory, and good knowledge levels chosen? Is this based on prior literature? Why not divide the participants into tertiles of STI knowledge scores, for example?

Please provide additional details regarding the measures collected in the questionnaire (and presented later in Table 4). For example, does “sexual partners” mean number of sexual partners in a specific time frame (eg, past 1 month?). What does “past relationships” mean – is this number of prior lifetime sexual partners or something else? Is condom use and sexual intercourse per week only asked of/reported for participants who report a current/recent sexual partner, or everyone? How is “heard of STI” asked or assessed? Is alcohol use any alcohol use at all, or over a certain amount, and in what time frame? Consider including a copy of the questionnaire as a supplementary file.

Line 160 – was a p-value threshold of <0.05 from the bivariate analyses used to determine if a variable would be included in the multivariable logistic regression, or a different threshold?

Results

Line 170 – Please define the acronym OPD for readers who may not be familiar with this.

Table 2 – Consider simplifying the table by removing or combining the cells for symptoms that are not applicable to one of the genders (eg, discharge from penis for female participants)

Lines 180-181 – The statement “low levels of knowledge did not translate to higher number of infections” seems like a conclusion to be made from a statistical analysis demonstrating that there was no association between level of knowledge and STI prevalence; it would thus be clearer to make this statement later in the results section where the univariate analyses are presented.

**Table 3 and lines 179-184 - Since there were 7 participants with co-infections, the total number of individuals with any STI (one or more) cannot be the same as the sum of the individual STIs. In looking at the number and type of coinfections described in lines 181-184, I have calculated out the total number of individuals with any STI to be 42 rather than 51, giving a different overall prevalence of any STI in the sample (42/178 = 23.6%). Please confirm if this is correct and amend the results accordingly. Additionally, note should be made in table 3 that individuals with co-infections are being counted multiple times in the table.

Line 188 – Age does not show a statistically significant association with having an STI based on table 4, instead, smoking status and number of past relationships do and should be included in this sentence instead of age.

Line 211 – please define ‘orthodox treatment’

Discussion

Line 219-220: It is unclear why the authors have compared the STI prevalence found in this study with STI prevalence among a different population (pregnant women) in a different part of the world (Nepal). This is not a relevant comparison, as there is no reason to suspect that these two prevalence rates should be similar. There are many studies reporting STI prevalence in more similar populations (symptomatic adults of both genders in West Africa) that would be a more appropriate comparison to the present study.

Lines 222 – 223: these percentages represent percent of the total cohort, not percent of the total infections.

Lines 223 – 229: were the Banong-le et al and Nyarko et al studies conducted among symptomatic patients or those attending STI services? Would be helpful to mention this if the prevalence rates are being compared.

Lines 236-238: is diagnostic testing available in Ghana? While increasing diagnostic testing would help avoid both over- and under-treatment of STIs, this is not available in many parts of the world. An acknowledgement and/or discussion of this challenge would strengthen the discussion section overall.

LInes 240-243: There is an important opportunity here to discuss the challenges faced by individuals in preventing STIs despite high levels of knowledge. Many factors have been identified, including social norms that discourage condom use, gender power imbalance, inability for women to negotiate condom use/safe sex, etc.

Lines 249-250: It would be helpful to mention that there were only 7 smokers total in the whole cohort and that the 95% confidence interval for the odds of STI for smokers vs non-smokers was very wide in multivariable analysis (1.07-39.56), therefore, these results have to be interpreted with caution, and it is possible that the risk of STI is only slightly higher for smokers vs non-smokers.

Lines 250-254: As above, I would be cautious with interpretation of this finding and with the results of the literature. In the Berg et al study, smoking was “associated with” higher risk sexual behaviors, it cannot be concluded that smoking itself “enhanced” a person’s tendency for higher risk sexual behavior.

Lines 261-262: have any studies (behavioral science, qualitative studies) reported on the cultural sensitiveness of discussing genitourinary issues in the Ghanaian setting? It would be helpful to cite literature from Ghana or a similar setting to support this statement.

Lines 263-264: did the study participants actually report that they had low confidence in the overall health system? If not, I would be clear here that this is only a possible explanation for the observed delays in seeking care.

Lines 278-279: the statement regarding incorporation of herbal medicines into the national treatment guidelines, while important, seems contradictory to the point made in the previous sentence regarding lack of evidence on efficacy and safety of herbal medicines. Perhaps the need is for greater research on the safety and efficacy of herbal medicines so that those that are safe and efficacious could be incorporated into treatment guidelines.

Conclusion

Line 295 – as only associations can be drawn, I would state that marital status and smoking “were associated with” (rather than “influenced”) STI risk

As STI knowledge was a major focus of the study, I would consider stating the finding that low STI knowledge was not associated with having an STI in the conclusions section.

Reviewer #5: The Authors have sufficiently addressed the concerns raised by the previous reviewers

7. PLOS authors have the option to publish the peer review history of their article (what does this mean?). If published, this will include your full peer review and any attached files.

**Do you want your identity to be public for this peer review?** For information about this choice, including consent withdrawal, please see our Privacy Policy.

Reviewer #4: No

Reviewer #5: No

[NOTE: If reviewer comments were submitted as an attachment file, they will be attached to this email and accessible via the submission site. Please log into your account, locate the manuscript record, and check for the action link "View Attachments". If this link does not appear, t

---

## [Editor Report · Decision Letter 2]

3 May 2024

PGPH-D-23-01356R2

Knowledge and prevalence of common sexually transmitted infections among patients seeking care at selected health facilities in Southern Ghana

Dear Dr. Hutton-Nyameaye,

Thank you for submitting your manuscript to PLOS Global Public Health. After careful consideration, we feel that it has merit but does not fully meet PLOS Global Public Health’s publication criteria as it currently stands. Therefore, we invite you to submit a revised version of the manuscript that addresses the points raised during the review process.

In particular, please address the comments below regarding the accurate reporting of STI prevalence rates. We encourage consultation with a biostatistician as needed. Additionally, please note the recommended limitations to address. 

We look forward to receiving your revised manuscript.

Kind regards,

Jana Jarolimova, M.D.

Guest Editor

Journal Requirements:

Additional Editor Comments (if provided):

1) The prevalence rates reported in the results do not reflect the true prevalence of each STI, as those STIs that were part of co-infections are not counted. Unless there is a reason to treat STIs that are detected as monoinfections vs co-infections differently (this would be unlikely), recommend that the prevelance rate for each STI is reported as the total number of people with each STI (including those with mono-infections and those with co-infections) dividided by the total sample size. of note, the sum of these prevalence rates will not be the same as the prevalence rate for any STI, as people with coinfections will only be counted once for the overall STI rate. 

2) Methods - clarify in lines 128-129 whether past relationships were lifetime relationships or relationships in past year.

3) Specify which answers on the STI knowledge survey were considered 'correct' (in the supplemental data)

4) Add to limitations:

    -Non-validated STI knowledge assessment used, may not be generalizable

    -Difficult to make associations between STI knowledge and STI prevalence among those already symptomatic - even those with low knowledge are arriving bc they’re seeking care. There could still be an association between STI knowledge and STI prevalence among people with asymptomatic infections.  
---

## [Editor Report · Decision Letter 3]

10 Jun 2024

Knowledge and prevalence of common sexually transmitted infections among patients seeking care at selected health facilities in Southern Ghana

PGPH-D-23-01356R3

Dear Dr Hutton-Nyameaye,

We are pleased to inform you that your manuscript 'Knowledge and prevalence of common sexually transmitted infections among patients seeking care at selected health facilities in Southern Ghana' has been provisionally accepted for publication in PLOS Global Public Health.

Best regards,

Jana Jarolimova, M.D.

Guest Editor

Reviewer Comments (if any, and for reference):

Please edit the last sentence in Results paragraph titled, "Prevalence and Knowledge of STIs among study participants" (lines 191-192), as the way that the text is currently worded, the total number of participants with coinfections would be 7 rather than 6. Specifically, the text states "Among the participants, 3.37% (n=6) had co-infections with two or three pathogens. Co-infections involving gonorrhoea and chlamydia were 2.25% (n=4). Other dual co-infections as well as coinfections with all three pathogens were recorded in one participant each." This would add up to: 4 patients with gonorrhea-chlamydia coinfection + 1 patient with gonorrhea-syphilis coinfection + 1 patient with chlamydia-syphilis coinfection + 1 patient with all three infections = 7 patients. however the true number should be 6.